# Recent Advances in the Diagnosis of Enamel Cracks: A Narrative Review

**DOI:** 10.3390/diagnostics12082027

**Published:** 2022-08-22

**Authors:** Bassam Zidane

**Affiliations:** Restorative Dentistry Department, King Abdulaziz University, P.O. Box 80209, Jeddah 21589, Saudi Arabia; bzidane@kau.edu.sa

**Keywords:** cusp failure, cracked tooth, diagnosis, enamel cracks, imaging, transillumination, swept-source optical coherence Tomography

## Abstract

Cracked teeth can pose a diagnostic dilemma for a clinician as they can mimic several other conditions. The constant physiological stress along with any pathological strain like trauma or iatrogenic causes can lead to the development of microcracks in the teeth. Constant exposure to immense stress can cause the progression of these often-undiagnosed tooth cracks to cause tooth fractures. This review aims to outline the etiology of tooth cracks, their classification, and recent advances in the diagnosis of enamel cracks. Diagnosing a cracked tooth can be an arduous task as symptoms differ according to the location and extension of the incomplete fracture. Early detection is critical because restorative treatment can prevent fracture propagation, microleakage, pulpal or periodontal tissue involvement, and catastrophic cusp failure. Older methods of crack detection are not sensitive or specific. They include clinical examination, visual inspection, exploratory excavation, and percussion test. The dye test used blue or gentian violet stains to highlight fracture lines. Modern methods include transillumination, optical coherence tomography Swept-Source Optical Coherence Tomography (SSOCT), near-infrared imaging, ultrasonic system, infrared thermography, and near-infrared laser. These methods appear to be more efficacious than traditional clinical dental imaging techniques in detecting longitudinal tooth cracks. Clinically distinguishing between the various types of cracks can be difficult with patient-reported signs and symptoms varying according to the location and extension of the incomplete fracture. Cracks are more common in restored teeth. Technological advances such as transillumination allow for early detection and enhanced prognosis.

## 1. Introduction

A cracked tooth refers to a partial enamel rupture in a living tooth with dentinal and even pulpal involvement [1]. The causative factors of a tooth crack vary. A masticatory function is the most common factor related to vertical tooth fracture [2]. Depending on the crack’s extension depth and subsequent bacterial infections, cracked teeth can cause several clinical symptoms [3]. Typical symptoms include acute pain on chewing or unexplained pain as a result of exposure to a cold stimulus [4]. An incidence rate of 34–74% has been reported in adults between the ages of 30–50 [5], with a female predisposition [6,7]. The probability of a cracked tooth is higher in an aging population with a greater number of retained teeth [8]. There is an increase in the mineral density of enamel with aging. This is concurrent with a decrease in the volume of the protein matrix [9]. Cosmetic procedures such as bleaching can accelerate this deterioration in fracture toughness and mechanical properties of the enamel [10,11]. 

The mandibular molars are more susceptible to enamel fractures. Fractured enamel is more prevalent in the upper and lower bicuspids as well as the upper molar teeth [7]. The mesiopalatal cusp of the upper first grinder is quite protuberant. This creates wedging on the first mandibular grinders, which might also contribute to tooth cracks [12]. As a consequence, these teeth are more liable to fracture. Diagnosing a cracked tooth can be an arduous task for a clinician as its symptoms can be variable and bizarre [13]. The cracks will continue to spread if prompt treatment is not given. This can eventually lead to pulpitis or a complete tooth fracture [14]. The primary objective of its diagnosis is early detection. Visual inspection [15], periodontal probing, bite diagnosis, pulp vitality test [16], staining [17], transillumination [18], and computed tomography (CT) [19] were the most common clinical diagnosis methods. Surface crack detection methods such as optical coherence tomography (OCT) have recently attracted a lot of attention [20,21]. Cone-beam computed tomography (CBCT) and Micro-CT appeared to be more efficacious than traditional clinical dental imaging techniques in detecting longitudinal tooth cracks [22,23]. This paper aims to outline the cracked tooth syndrome, diagnostic methods used over the years, and the recent advances in diagnosis.

## 2. Clinical Features

The clinical symptoms differ according to the location and extension of the incomplete fracture [12,24,25]. It is possible to elicit a history of months of discomfort and acute pain while chewing or consuming cold beverages. “Rebound pain” or pain that occurs when pressure is released after eating fibrous foods is frequently observed. Patients may not be able to identify the tooth where the pain is felt. They may report a negative response to heat stimuli. 

Microleakage of bacterial byproducts and toxins can cause chronic pulpitis without causing clinical symptoms. Pulpal and periodontal symptoms may develop from cracks with pulpal involvement [26,27].

The physiological basis of chewing pain as proposed by Brannstrom and Astrom [28] states that the independent movement of the fractured tooth part with each other could cause a sudden transition in the fluids within the dentinal tubules. Subsequently, the A-type fibers get activated in the dental pulp leading to immediate pain. The toxic irritants may leak via the cracks producing hypersensitivity to cold. The release of neuropeptides as a result of the toxic irritant leakage within the dental pulp leads to a decreased pain threshold. These symptoms occur because of alternate compression and stretching of the odontoblast processes inside the crack.

Asymptomatic enamel cracks can have multiple pathogenic consequences. Teeth with enamel cracks may be further damaged by ultrasonic instrumentation such as scaling and root planing [29]. 

## 3. Difference between a Fractured Cusp or a Split Tooth and a Cracked Tooth

A cracked tooth can be differentiated from a split tooth by checking the movements of the segment by wedging. A cracked tooth demonstrates no movement with wedging pressures. Under light pressure, a split tooth may break off, leaving patients with no further mobility. Under wedging stresses, a fractured cusp will display mobility and the mobile part will extend deep beyond the cementoenamel junction. Postoperative sensitivity as a result of broken restoration, areas of hyper occlusion due to dental restoration, microleakage from newly filled composite resin restorations, orofacial pain, and bruxism pain, can all be considered as a cracked tooth [30].

## 4. Causes of Enamel Cracks and Identification

The sources and causes of enamel cracks are multifactorial. They can develop glacially over a long time period. Enamel cracks can exist independent of dentin cracks. The pain from a cracked tooth can be difficult to distinguish from other conditions such as ear pain, atypical orofacial pain, migraine, sinusitis, or temporomandibular joint disorders. As a consequence, diagnosis may be clinically challenging and delayed. Early detection is critical because restorative treatment can prevent fracture propagation, microleakage, pulpal or periodontal tissue involvement, and catastrophic cusp failure [31].

The ease at which a fracture can be diagnosed varies depending on its location and extent. An extensive intracoronal repair is usually present on the tooth. Humans uniquely consume both hot and cold food, leading to repeated thermal changes that can cause expansion and contraction [32]. This is particularly relevant in the case of teeth restored with ceramics where thermal expansion and cyclic loading can lead to crack formation and propagation. Microscopic cracks at the edges of a restoration can indicate structural weakness [18]. Cracks at the margins of composite restorations may be due to polymerization contraction stress. It can present as a whitish margin [33]. Well-defined discoloration of a cusp can be a sign of a lack of structural integrity [18]. Pain can arise after dental procedures, such as the cementation of an inlay, and is often misdiagnosed to be due to the high spots on the recent restoration. The presence of underlying cracks could be indicated by regularly debonding the cemented intracoronal restorations like inlays.

## 5. Classification/Definitions of Tooth Fracture 

The American Association of Endodontists has divided longitudinal tooth fractures into five classes: craze line, fractured cusp, cracked tooth, split tooth, and vertical root fracture [34]. An asymptomatic crack in the tooth enamel may progress along with dentin cracks. Clinically distinguishing between the various types of cracks can be difficult (Figure 1). 

Cracked tooth syndrome includes three phases:Craze Lines: Tiny and harmless cracks that occur in the outer enamel of the tooth.Cracks: It occurs when craze lines penetrate the body of the tooth (dentin).Fractures: A fracture occurs when cracks expand deep into the root of the tooth.

Craze lines are asymptomatic superficial lines that appear on enamel with age. They are frequently confused with cracks but can be distinguished with transillumination. Fractured cusps initiate from the coronal tooth portion, progress into the dentinal area, and end cervically. They are generally connected with large restorations that leave the cuspal enamel unsupported. A cracked tooth is defined as one that extends apically from the crown of the tooth without separating two segments, according to the AAE. A split tooth has a breakthrough both marginal ridges, totally dividing the tooth into two halves. Vertical root fractures begin from the root and are usually complete or partial. All classification methods have the difficulty of failing to relate the descriptions to clinical outcomes or treatment recommendations.

A tooth fracture was divided into two types by Silvestri and Singh (1978): complete and incompletely fractured teeth [35]. Complete fractures were then separated into vertically directed and obliquely directed complete fractures. The authors believe that the full oblique fracture occurs most frequently as a result of a thorough restoration that undermines a cusp. The undermined cusp fractures off under the stresses of physiological functions. A vertically oriented complete fracture is seen as two individual mobile tooth and root segments in relation to one another. The incomplete tooth fractures are of two types: vertical and oblique. An oblique crack originates on the occlusal surface of the tooth, involves one or few cusps, extends through the dentin in an oblique direction beneath the cusps, and ends gingivally in the cementum or enamel. Tooth segments are not completely crushed. Vertical incomplete fractures start in the enamel and extend through the dentin and the root progress. With no complete separation of segments, the crack might involve one or both the marginal ridges extending in the mesiodistal direction. Figure 2 depicts a more detailed classification scheme, and was created by Talim and Gohil (1974): [36]. 

## 6. Early Methods of Diagnosing Enamel Crack

### 6.1. Clinical Examination 

Careful clinical examination can visualize localized periodontal defects (subgingival cracks), facets on the occlusal surface of the tooth, and positive response to thermal or sweet stimuli. Once the tooth has been located, many authors recommend removing old restorations and stains to view the cracks. Rubber dam usage has the advantage of detecting the cracks more easily by keeping the tooth surface free of saliva, highlighting the crack with a clear background and preventing the interference of surrounding structures. 

### 6.2. Visual Inspection 

Visual inspection is inherently limited. Traditional visualization may not be sensitive enough to assess the presence or severity of the cracks. Although visual examination of the tooth is beneficial, cracks cannot be visualized without magnifying loupes. While it can be identified, it is not always obvious. Once the tooth has been located, many authors recommend removing old restorations and stains to aid in the visualization of the crack [37,38,39,40,41].

### 6.3. Exploratory Excavation

It is often difficult to find the crack beneath the removed restoration, therefore the tooth should be further excavated with the consent of the patient. 

### 6.4. Percussion Test

The cracked tooth could be diagnosed with a positive response to apical percussion. 

### 6.5. Periodontal Probing

The cracked tooth can be differentiated from the split tooth using periodontal probing when the line of fracture progresses beyond the gingiva. Careful probing of suspected cracks is required to reveal a periodontal pocket. Deep probing, conversely, frequently helps in the diagnosis of a split tooth, which indicates a poor prognosis.

### 6.6. Dye Test

Fracture lines can be highlighted with methylene blue or gentian violet stains [42]. This is due to the dye’s tendency for aggregation. However, using the dye may hide cracks or cause slight color changes in the enamel’s deeper layers [18]. In addition, the original restorative elements must be excavated before dye application, which takes 2–5 days [43]. Another problem is that a permanent aesthetic restoration is not feasible.

### 6.7. Transillumination 

Transillumination is a useful tool for locating tooth cracks [44]. The tooth must be cleansed before transillumination, and the light source must be held above the tooth. The presence of any crack penetrating the tooth dentin will derange the transmission of light [45]. For traditional crack diagnosis, transillumination is the frequent method used (Figure 3). The use of transillumination without magnification has two disadvantages. Transillumination, for starters, causes the craze lines to give the impression of physical fractures. Additionally, the minor color shift is undetectable. The use of magnification and fiber-optic transillumination will aid in the visualization of a crack. The detected cracks, do not precisely signify the extent and form of the cracks [46].

The intensification to determine its spectrum averages from 14–18×. As per experienced clinicians, 16 times seems to be the ideal magnification [47]. The minor cracks can be properly detected using the fiber-optic transillumination equipment and the dental microscope. These are widely used as effective diagnostic tools in clinical practice.

Transillumination combined with methylene blue allows better discrimination of enamel cracks than either modality alone [40].

### 6.8. Bite Tests

Bite testing is performed using cotton wool rolls, orangewood sticks, rubber coarse rolls, or a 10-number round bur in a cellophane tape grip. The subject is instructed to close his mouth on an individual cusp while using wooden sticks, thereby aiding fracture cusp isolation. Cracks can be detected using cotton rolls. The subject is asked to close his mouth on the roll before suddenly releasing the pressure. The presence of pain upon sudden pressure release confirms the diagnosis. Other devices like Tooth Slooth II (Professional Results Inc., Laguna Niguel, CA, USA) and Fractfinder (Denbur, Oak Brook, IL, USA) are said to have a higher level of sensitivity when compared to wooden sticks. These along with rubber plungers of numbing ampules adjourned on a dental floss are utilized in the same way as cotton rolls. Investigation is confirmed when there is aching on sudden pressure retrieval. It aids in the correct identification of the cusp in concern. The majority of vitality tests are positive. However, due to the existing pulpitis, the affected teeth show hypersensitivity to cold stimuli. This characteristic may establish confirmation of the cracked tooth. Other researchers however believe that putting pressure on the teeth with questionable symptoms may extend the crack propagation, and hence do not recommend bite testing [13].

### 6.9. Radiograph 

Radiographs can help to determine the overall tooth status, however, a crack is seldom seen on one [39,41,48,49]. Research has shown that conventional radiography such as bitewing imaging shows poor accuracy in crack detection [50]. Conventional periapical X-rays (PR) can be recommended when a root fracture is deviated [51], however, cone-beam computed tomography (CBCT) can identify minor periapical bone loss during vertical root fractures (VRFs) [52]. However, because CBCT has a resolution of just about 80 μm, the diagnosis of cracked teeth or detection of vertical root fractures becomes difficult [53,54].

Yuan et al. [55] showed in an in vitro study that a contrast, meglumine diatrizoate can improve scanning with CBCT as compared to the usual approach because it can effectively highlight the hidden cracks. As a result, this may be an additional method for imaging the periapex of the tooth [56].

## 7. Recent Advances in Diagnosing Enamel Cracks

### 7.1. Swept-Source Optical Coherence Tomography (SSOCT)

Optical coherence tomography is a cross-sectional diagnostic method that helps to envision optical characteristics of biological tissues. Its principle is identical to that of ultrasonic pulse-echo imaging, in which it utilizes infrared waves which reflect the internal microstructure [57,58]. Swept-source optical coherence tomography (SS-OCT) is highly sensitive [59,60,61]. It is an improvement over traditional CT for detecting cracked teeth and incipient enamel caries [62,63,64]. It allows for the construction of cross-sectional images of internal biologic structures [65]. It may also have applications in the diagnosis of restorative failures in the future [66,67]. SSOCT is a modification of the Fourier domain techniques that offer a discrete rise in sensitivity in comparison to the traditional optical coherence tomography techniques [68].

Literature suggests that a wavelength of light in the infrared region (700 to 1550 nm) shows tremendous scope for enamel imaging due to the mild absorption and scattering in this range of light [61,69]. SS-OCT uses a laser of suitable frequency to emit multiple wavelengths of light (near-infrared wavelength of 1300 nm [70]. Using SS-OCT, the enamel is highly transparent and any cracks present are revealed with good contrast [71]. With the help of a laser scanning device and a semiconductor camera, this imaging approach uses low-consistent interferometry for capturing the reflected waves of biostructures at varying pits facing weak coherent light. It creates two or three-dimensional mechanical structures [72,73]. SS-OCT has higher diagnostic accuracy than microCT, FOTI, and visual inspection [20,66]. 

Although SS-OCT increases resolution, it shows low specificity for sensing full-thickness cracks as the improved image of deep fractures frequently intersects the enamel plexus. Cracks extending beyond the amelodentinal junction can also be imaged using the SS-OTC, thus serving in the evaluation of the crack depth. The coronal section of the SS-OCT shows a narrowed abyss of penetration of 3 mm that can be laser irradiated. As a result, its primary applications are limited to confirmation of the initial investigation [21,66]. 

SS-OCT pictures reveal fracture lines and mechanical fractures. It can detect vertical fractures in extracted human teeth [74,75]. However, this detection is restricted only to the coronal area where the laser beam is irradiated. Its detection abilities are comparable to micro-CT (100%) and transillumination [20]. It can detect a whole-thickness enamel crack with the location of the crack showing light backscattering. Enamel thickness varies at different sections in a tooth. This non-uniform composition and the surface roughness have an influence on penetration depth and image contrast of SS-OCT. With the Swept-Source Optical Coherence Tomography, a longer wavelength may be used to increase the penetration dep. However, this lowers the axial resolution in the cross-section which in turn causes a struggle with crack detection [73]. 

### 7.2. Near-Infrared Imaging

Near-infrared transillumination is a recent diagnostic modality for enamel cracks [73,76,77]. Imaging with the long wavelengths of 1300–1700 nm of near-infrared illumination gives the greatest contrast between enamel and dentine [78]. Near-infrared imaging is capable of detecting of several cracks. However, it does not provide information about the precise depth of the crack nor can it differentiate between the different types of cracks [21,44]. This technique was developed to overcome the limitation of insufficient scanning range of the Swept-Source Optical Coherence Tomography [79,80]. In the Swept-Source Optical Coherence Tomography, a whole lesion scan is time-consuming and requires the patient to be immobilized during the scanning. Any movement can hamper the image quality and lead to motion-related artifacts [81,82]. Near-infrared imaging has the advantage of rapid screening of the disease region over the Swept-Source Optical Coherence Tomography 

Near-infrared fluorescence (ICG-NIRF) dental imaging offers a solution for detecting cracks extending into enamel or dentine [83]. While CT scans can reveal dentin cracks, NIR images can reveal most enamel flaws that are invisible on X-ray imaging. The direction of light is a key component in the detection of a fracture: an angled exposure produced better image contrast of cracks than a parallel exposure because a shadow is produced beneath the line. The line shadow in ICG-NIRF and NIRi-II images could be used to measure the depth of the line and differentiate dentino-enamel cracks from craze lines using this shadow. ICG-NIRF images with 1-min ICG tooth immersion revealed cracks clearly, though extended ICG immersion gave images better contrast. 

The quantitative light-induced fluorescence (QLF) quantifies the percentage of fluorescence change of demineralized enamel in comparison to the adjacent enamel. It is related directly to the quantity of mineral lost at the time of demineralization [84]. Jun MK examined the effectiveness of quantitative light-induced fluorescence (QLF) technology in the identification of enamel cracks on 96 human extracted teeth using blue light of wavelength 405 nm. This light is known to approach dentine beneath the enamel causing tooth fluorescence. In case of an existing lesion, some light gets shielded and portrayed as a dark area [85]. A strong correlation was seen between the histologic evaluation and the maximum fluorescence lost (F max) value, with a correlation coefficient of 0.84. It was found that for enamel inner-half cracks, QLF-D had a sensitivity of 0.87 and a specificity of 0.98, whereas cracks entering the dentino-enamel junction had a sensitivity of 0.90 and a specificity of 1.0. This suggests that QLF technology could be a valuable clinical tool in identifying enamel defects, particularly because it is nondestructive. 

### 7.3. Cone-Beam Computed Tomography (CBCT)

Contrast-enhanced cone-beam computed tomography (CBCT) was utilized by Zhou J et al. [86] to enhance the accuracy of detection and determination of the depth of the crack. Contrast-enhanced CBCT under vacuum conditions is known to suggestively improve the diagnostic rate of cracks. However, it is not a very accurate measure of crack depths. 

### 7.4. Other Techniques 

a.Ultrasonic System

Detection of CTS using ultrasonics has a promising future as it demonstrates the capability to infiltrate hard tissues, including radiopaque restorations. It is also oblivious to the hazards of ionizing radioactivity. Culijat et al. [87] used a system of integrated ultrasonics to successfully identify known cracks in simulated teeth. The first study to use laser ultrasonics utilized image analysis in combination with analysis of finite-element to precisely measure the crack depth in clinical settings was published by Sun et al. [86].

b.Infrared Thermography

When other diagnostic techniques fail, infrared thermography technology can assist in the detection of small cracks (4–35.5 m). The ultrasonic power vibrates (the amplitude and detection angle should be 0.89 W within 45°, respectively) and creates local friction that generates heat [88,89]. The dentinal microcracks can then be presented under the action of the thermal imager. However, this approach has several shortcomings when it comes to detecting wide cracks [90].

c.Near-Infrared

A near-infrared diode laser with a wavelength of 810 nm can be employed as new technology to help control systematic CTS. When teeth in question are exposed to laser energy, the majority of patients feel a shooting ache, with a handful feeling a dull ache. This can be attributed to the energy applied to the pulp when the laser beam enters the depth of the crack generating an analogous irritation [36]. The classical investigation techniques and new advancements in clinical diagnosis should focus on problems that arise earlier during the procedure. 

Zheng Y et al. examined an in vivo system that used a near-IR light source to identify enamel cracks and analyze an association between age and anterior enamel cracks [91]. Qualitative examination revealed no correlation between age and the gravity of the enamel cracks. Although not statistically significant, a tendency to increase the length of anterior enamel cracks was seen with age. An 850 nm wavelength silicon charge-coupled device (CCD) shows satisfactory results in detecting enamel cracks. 

Chen Zhi et al. evaluated four distinct methods for detecting cracks in teeth, which may elicit foundations for timely crack detection [92]. Three observers independently examined crown cracks using the naked eye, microscope, methylene blue dye, and methylene blue dye with magnification on 123 freshly extracted human teeth. It was concluded that methylene blue dye used under magnification detected tooth cracks with greater accuracy. The detection of the cracks also depended on the experience and diagnostic skills of the endodontist. Table 1 lists the common methods for identifying cracked teeth.

## 8. Conclusions

Patients with a cracked tooth may present with a confusing array of symptoms. A cracked tooth is a distinct possibility to consider when with complaints of pain or discomfort while chewing. Enamel cracks are ominous harbingers of future pathology and need to be treated promptly. Current conventional diagnostic methods may be inconclusive. Dental professionals with a keen understanding of the classic signs combined with recent technological advances can help in early diagnosis and prevent further crack propagation and destruction.

## Figures and Tables

**Figure 1 diagnostics-12-02027-f001:**
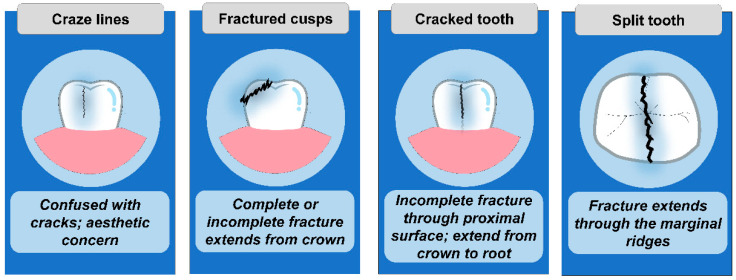
Type of tooth fractures.

**Figure 2 diagnostics-12-02027-f002:**
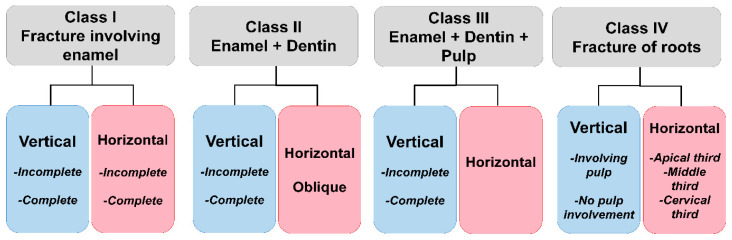
Classification of tooth fractures [36].

**Figure 3 diagnostics-12-02027-f003:**
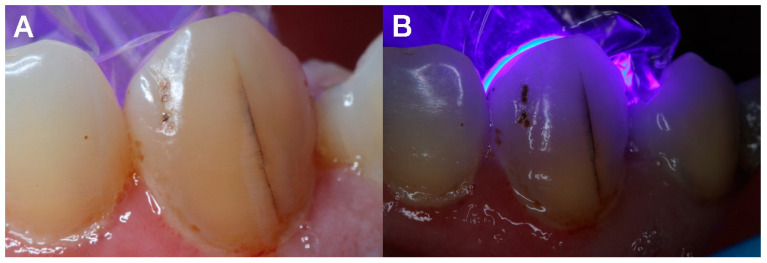
(**A**) Enamel crack. (**B**) Transillumination used to confirm presence of crack.

**Table 1 diagnostics-12-02027-t001:** Four common methods for identifying cracked teeth [13].

Features	Transillumination	Intraoral X-ray	CBCT	SS-OCT
Distinguish the type of crack	× [70]	× [90]	× [18]	✓ [86]
Show root fractures	× [70]	✓ [1]	✓ [81]	× [86]
Determine the crack depth	× [18]	× [10]	✓ [86]	✓ [17]
Produce radiation	× [92]	✓ [50]	✓ [56]	× [18]

## Data Availability

Not applicable.

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
