# Peer review of "Recent Advances in the Diagnosis of Enamel Cracks: A Narrative Review"

_diagnostics, 2022, doi:10.3390/diagnostics12082027_

Round 1

Reviewer 1 Report

Very comprehensive and well written review. No corrections from my side

Author Response

thanks for your in-put

Reviewer 2 Report

The paper in question is presented as an update of a particular Topic the Enamel Crack. The term Review is often used in the text, for this reason authors should decide whether it's an update or a Review.  If it's a Review it's good that they indicate what type they conducted and this metodology must be evident in the text. Otherwise if it's an update on a pathological theme without the limitis and characteristics of the Review. The Narrative Reviews have very similars characteristics to the proposed paper, please the Author to clarify this aspect.

Author Response

this article is a review article for the Enamel Crack topic, an updated version has been uploaded 

Reviewer 3 Report

Dear Author 

1) Abstract is well written. 

2) Keywords are not well proposed. Revised them. 

3) Introduction: all the written part is from published work. Where is the author own points of view? Try to revise it and write your own words. for exam this statement "The mandibular molars are more susceptible to enamel fractures" What is the cause or reason for mandibular molar enamel fracture? 

4) Heading 2: try to add some radiographic images representing enamel cracks or any histological picture from published work. 

5) Figure 1 and 2 require reference or source of designing. 

6) In heading number 4: author not discuss about the diagnosis of enamel cracks or tooth fracture. Poor knowledge of scientific information. Its a serious topic and author misleading the heading with out of context. 

7) same issue in the heading number 5. Authors not prepared well look like some student assignment. If author raisng a recent advances about the topic then why they forget to discuss very important society of dental trama guidelines? https://www.iadt-dentaltrauma.org/for-professionals.html 

8) Heading number 7: very poor writing and lack of information. Author are not using his own statement everything look like copy and rephrase. quite poor clinical evidence synthesis. 

9) conclusion is poor. 

8) 

Author Response

Response to Reviewer’s comments

Reviewer’s comment

Author’s response

Reviewer 1

Very comprehensive and well-written review. No corrections from my side.

We are grateful for the time taken by the Reviewer in going through our manuscript and providing us with helpful comments.

Reviewer 2

The paper in question is presented as an update of a particular Topic the Enamel Crack. The term Review is often used in the text, for this reason authors should decide whether it's an update or a Review.  If it's a Review it's good that they indicate what type they conducted and this methodology must be evident in the text. Otherwise if it's an update on a pathological theme without the limits and characteristics of the Review. The Narrative Reviews have very similar characteristics to the proposed paper, please ask the Author to clarify this aspect.

We are grateful for the time taken by the Reviewer in going through our manuscript and providing us with helpful comments.

We thank the reviewer for their keen insights and providing us invaluable feedback to improve our manuscript.

The title has been changed to a narrative review

Reviewer 3

1) Abstract is well written. 

 We thank the reviewers for their kind comments

2) Keywords are not well proposed. Revised them. 

 Keywords have been revised

3) Introduction: all the written part is from published work. Where is the author own points of view? Try to revise it and write your own words. for exam this statement "The mandibular molars are more susceptible to enamel fractures" What is the cause or reason for mandibular molar enamel fracture? 

 The mesiopalatal cusp of the upper first grinder is quite protuberant. This creates wedging on the first mandibular grinders, which might also contribute to tooth cracks.[12] As a consequence, these teeth are more liable to fracture

4) Heading 2: try to add some radiographic images representing enamel cracks or any histological picture from published work. 

5) Figure 1 and 2 require reference or source of designing. 

 Both figures were created by the author

Figure 2 is based on the classification of tooth fracture by Talim and Gohil, 1974.

6) In heading number 4: author not discuss about the diagnosis of enamel cracks or tooth fracture. Poor knowledge of scientific information. Its a serious topic and author misleading the heading with out of context. 

 The heading has been revised

7) same issue in the heading number 5. Authors not prepared well look like some student assignment. If author raisng a recent advances about the topic then why they forget to discuss very important society of dental trama guidelines? https://www.iadt-dentaltrauma.org/for-professionals.html 

 International Association of Dental Traumatology Guidelines for the Management of Traumatic Dental Injuries did not describe guidelines for enamel cracks

8) Heading number 7: very poor writing and lack of information. Author are not using his own statement everything look like copy and rephrase. quite poor clinical evidence synthesis. 

 Heading 7 has been revised and the text has been

9) conclusion is poor. 

 The conclusion has been rewritten

Round 2

Reviewer 3 Report

Dear Authors 

I recommend to add some radiograph or clinical images. This is a clinical dentistry paper. 

Author Response

all the points were addressed in the newer version 
